# Binder-type effect on the physico-mechanical, combustion and emission properties of *Alstonia boonei* De Wild. sawdust and *Theobroma cacao* L. pod biochar briquettes for energy applications

**Mark Glalah**[1]*, **Charles Antwi-Boasiako**[2], **Derrick Adu-Gyamfi**[2]

**1** Department of Wood Technology, Faculty of Built and Natural Environment, Tamale Technical University, Tamale, Ghana, **2** Department of Wood Science and Technology, Faculty of Renewable Natural Resources, Kwame Nkrumah University of Science and Technology, Kumasi, Ghana

☯ These authors contributed equally to this work.
* markglalah@gmail.com, mglalah@tatu.edu.gh

**Data Availability Statement:** The study's underlying data set is accessible through figshare.

## Abstract

Energy application potential from the abundant biomass residues is inadequately exploited. Over-dependence on forest trees, its negative environmental impacts, and ever-rising energy costs require alternative production technologies including briquetting. The physico-mechanical and combustion properties of binderless and bindered *Alstonia boonei* sawdust and *Theobroma cacao* (cocoa) pod briquettes, carbonized in a steel kiln (at 410±5˚C, and a heating rate of 4˚C/min from the ambient temperature of 25˚C), piston-pressed at 9.0 MPa, were studied. The binders were starch, wax, and clay. Starch-bindered *T. cacao* pod briquettes recorded the maximum bulk density (640 kg/cm$^3$), while basic density was greatest for sawdust/clay briquette (433 kg/cm$^3$). Sawdust/wax briquette produced much Water Resistance Capacity (76.76%) with safer carbon monoxide (CO) emissions (0.67 ppm). *A. boonei* sawdust/starch briquettes recorded the greatest calorific value (24.023 MJ/kg), least specific fuel consumption (0.0483 kg/l), and slowest burning rate (0.0005 kg/min). All but *T. cacao* pod/starch and Sawdust/starch emitted CO below the safe air quality Standard of ≤ 6ppm (24h mean). Binderless sawdust, sawdust/starch and *T. cacao* pod/starch briquettes recorded 47.86, 20.95 and 11.40 μg/m$^3$ particulate matter (PM$_{2.5}$) respectively, which are below WHO Air Quality Standard safe for domestic uses. Binderless *T. cacao* pod produced more harmful CO and PM$_{2.5}$ than its non-bindered *A. boonei* sawdust counterpart. Clay-bindered briquettes were the most durable. Briquetting, 'a waste-to-energy technology', enhances bio-residue management for domestic and industrial spaces in the global energy mix.

## 1. Introduction

The indiscriminate felling of trees for wood fuel, significantly contributes to the reduction of our dwindling forest resources. Energy demand has increased considerably due to population

Link: https://doi.org/10.6084/m9.figshare.24408217.v2.

**Funding:** The authors received no specific funding for this work.

**Competing interests:** The authors have declared that no competing interests exist.

growth and urbanization [1]. The increase, however, is most pronounced in wood fuel consumption, particularly charcoal [2]. Wood fuel production and use are highly noticeable in tropical countries [3]. Asia accounts for about 44% of all wood fuel use, about 21% in Africa, while South America and the Caribbean use about 12% [4]. Large quantities of forestry- and agricultural-based residues are generated annually. These could be engineered as alternate fuel sources whose use could greatly offset negative impacts of forest extraction for energy and environmental pollution. However, their utilization has become a challenge in the development context [5]. Sayigh [6] reported that over 33% of energy consumption for developing countries could be supplied from biomass resources, which contribute about 64% of Ghana's total primary energy; yet the resource is not fully utilized [7].

West Africa is widely noted for cocoa production, which contributes immensely to the economic growth of the Subregion. Ghana produced about 822,000 metric tons of cocoa in the 2021/2022 harvest season, which made her the second largest producer in the world [8]. Its pod husks, after the bean extraction, form over 70% of the matured fruit [9]. Ghana produces over 1.5 million tons annually, which go to "waste" [10]. However, when they are developed into commercially high-value products (e.g. biochar briquettes), they would contribute to the improvement of the national economy and provide sustainable livelihood for the inhabitants. Similarly, a great amount of wood residue are generated after timber harvesting and processing [11]. These are in the form of cut-offs, barks, slabs and sawdust. They account for 15–60% by volume of wood in sawmills in many developing countries [12]. Sekyere et al. [13] noted that the annual sawdust generated in only two Ghanaian mills (Logs and Lumber Ltd and Asuo Bamosadu Timbers and Sawmills Ltd) ranged from 19,230 to 32,610 tons. Kumasi Sokoban Wood Village alone generates about 150 tons of sawdust daily [14]. Most of these Timber Firms burn their wood-residues. The smoke causes air pollution, global warming and respiratory diseases. Biomass-residue conversion for energy using comtemporary technologies would curtail the threats to human health, and also reduce the large sums of money for their disposal.

These biomass residues generated by the wood- and agricultural-based industries in developing countries can replace the domestic demand for traditional energy sources such as firewood and charcoal [15]. With the present energy-driven technologies in place, it is estimated by the West African Clean Energy and Environment Exhibition (WACEE) [16] that Ghana has the potential to produce about 110 MW of power from biomass residues only.

Biomass is accepted globally among the best alternative energy sources; it is readily available, cheap, carbon neutral and supplies almost six times the combined energy generated by solar, geothermal and wind sources [17]. The emission of $CO_2$ from biomass combustion is equivalent to the amount absorbed during its growing cycle, so the net $CO_2$ released is approximately zero by mass [18]. Carbonized biochar briquettes are however more energy efficient than biomass, as the latter combusts with a greater CV and with a generally lower $CO_2$ emission.

Biomass residues can be transferred into other energy products including biochar briquettes whose production involves biomass residue pyrolysis, and compaction into a uniform solid fuel with or without binders. It is used for heating, cooking, barbequing, and camping sessions, as in cold areas of the globe [11]. The type of binders used in briquettes is very important as their constituents influence the energy and mechanical properties of briquettes [19]. The properties of biochar briquettes are interdependent and collectively influence their performance. The present work sought to investigate the effect of binder types (i.e., clay, starch, and wax) on the physical properties [i.e., bulk and basic densities, and Water Resistance Capacity (WRC)] and mechanical properties of *Theobroma cacao* L. (cocoa) (Fam. Malvaceae) pod husk and *Alstonia boonei* De Wild. (Fam. Apocynaceae) sawdust briquettes. Examined also were their combustion properties (i.e., Gas Emission Analysis, water boiling analysis, burning

rate and Specific Fuel Consumption) and durability characteristics (e.g. fracture toughness and shatter resistance). Biochar briquettes were analyzed over traditional briquettes because unlike the latter, charred briquettes have garnered significant attention in research and are more widely used. The former possesses a higher energy density compared to traditional briquettes, as they possess more fixed carbon content, which translates to more energy per unit of mass. This makes biochar briquettes more efficient as a fuel source, providing more heat and longer burn times. Additionally, biochar combustion also results in lower emissions of harmful pollutants, making them a cleaner and energy solution. Briquetting would enhance many micro- and macro-enterprise opportunities along its value chain (through production, packaging and transportation) to the local and international markets. Thus, this study is a contribution to the achievement of the UN Sustainable Development Goals (especially 7 and 12), which take into perspective affordable, clean energy and responsible production and consumption [20].

## 2. Materials and methods

### 2.1. Charring and sieving of dried T. cacao pod husk and A. boonei sawdust

*T. cacao* (cocoa) pods and *A. boonei* sawdust were collected from the Atwima Mponua District, and the Sokoban Wood Village (Kumasi, Ghana) respectively, and air-dried for 5 days before carbonization. Charring of the samples was carried out separately at the Food Processing Laboratory of the Technology Consultancy Center (TCC), Kwame Nkrumah University of Science and Technology (KNUST), Kumasi. For carbonization, 4kg of each dried biomass was loaded into a metallic kiln and the top closed with a metallic lid attached to a conical chimney to avoid oxygen inflow. The charge was fired and the doors closed tightly to start the pyrolytic process (at 410±5˚C, with a heating rate of 4˚C/min from the ambient temperature of 25˚C), After 45min. of ignition, the temperature was evaluated intermittently with an infrared thermometer. After carbonization, the charge was allowed to cool for 24 h. Each processed sample was stored in air-tight containers.

### 2.2. Biochar briquette production from T. cacao pod husk and A. boonei sawdust using different binders

Three binders were used for briquetting, namely starch (a commercial binder) $[C_6H_{10}O_5)_n$, a polysaccharide composed of amylose and amylopectin, MSDS (Material Safety Data Sheet) = pH: 5.0–7.0, reactive hazard: none known, acute toxicity: none known, harzardous polymerization: does not occur [21]] because of its relative availability, low cost, and ease of preparation, wax [a mixture mainly alkane (saturated hhydrocarbons) $C_nH_{2n+2}$, MSDS = odor: faint to mild, vapour pressure: 0.013 kPa at 20˚C, autoignition temperature: not defined, boiling point/range: >310˚C, hazardous polymerization: will not occur [22]] for its extreme stability and insolubility in water, and clay $[Al_2Si_2O_5(OH)_4$, typically composition of clay: silicon (Si), aluminum (Al), oxygen (O), hydrogen (H) (in the form of hydroxyl groups, OH), MSDS = odorless, pH: 3–7, exposure limits: N/A, reactivity: not reactive under ambient conditions, chemical stability; stable under ambient conditions, toxicological information: not toxic/ not classified [23]] for its least smoke emission [24]. The starch paste (i.e., 100 g with 800 ml of water) was boiled at 100˚C until it became sticky. The wax was similarly prepared (100 g with 800 ml of 100˚C boiled water), and the clay (100 g with 800 ml of water) without boiling (at ambient temperature). The carbonized cocoa pods and *A. boonei* sawdust were crushed into smaller particles and sieved using Tyler sieves to obtain uniform-sized particles (< 2mm) [25]. Each charred sample (600 g) was thoroughly mixed with the respective binders (100 g) to a uniform mixture (Table 1). For the pulverised carbonized cocoa pods and *A. boonei* sawdust,

**Table 1. Charred biomass type and binder for biochar briquette production.**

| Binder type | Biomass type | Briquette Composition |
|---|---|---|
| Clay | *T. cacao* pod husk | *T. cacao* pod/clay^ |
| Starch | | *T. cacao* pod/starch^ |
| Wax | | *T. cacao* pod/wax^ |
| No binder | | *T. cacao* pod* |
| Clay | *A. boonei* sawdust | *A. boonei* sawdust/clay^ |
| Starch | | *A. boonei* sawdust/starch^ |
| Wax | | *A. boonei* sawdust/wax^ |
| No binder | | *A. boonei* sawdust* |

^ implies biochar briquette composition = 6(biomass):1(binder) w/w

* implies no binder was used, and the composition = 6(biomass):0(binder) w/w, i.e., 100% biomass only

600g of each (without binder) was separately collected to mould their respective briquettes. Distilled water was gently added to each mixture and stirred for easy compaction into the briquettes. Each bindered mixture (50 g) was fed gradually into a cylindrical steel mould (14.3 cm high × 5 cm wide) and compacted with a Piston press (GMC-YKY012) at 9.0 MPa for 60 sec. Non-bindered charred samples (i.e., binderless briquettes) were the controls. Three briquette replicates of each type were produced for testing.

## 2.3. Physical properties of T. cacao pod husk and A. boonei sawdust biochar briquettes

**2.3.1 Moisture content.** The moisture contents of the *T. cacao* pods and *A. boonei* sawdust biochar briquettes and their controls (non-bindered) were determined using a Delmhorst Moisture Meter (J-2000).

**2.3.2. Basic and bulk densities.** For basic density, a known mass (45 g) of each biochar briquette sample was tightly wrapped in light cling film (Everpack®) to prevent water imbibition and gently immersed in 500 ml of water in a vessel. The mass of the water displaced by the piece gave it its volume. Each piece was oven-dried (at 105±2°C) for 24 h to a constant mass. Its basic density was obtained by dividing the mass by its original volume [25].

The bulk density of each sample was determined with a cylindrical container (1000 ml). The mass of the container was taken; it was filled with the sample and re-weighed. The bulk density was determined by dividing the mass of the biochar briquette with its volume [25, 26]:

$$\text{Bulk density} = \frac{Mass\ of\ biochar\ briquette\ sample\ (g)}{Volume\ of\ measuring\ cylinder\ (cm^3)} \tag{1}$$

**2.3.3. Water absorbance and resistance capacity.** The measure of the water absorbed (%) by a biochar briquette when immersed in the solvent was made. Each briquette (45 g) was immersed in 500 ml of water in a vessel at 27°C for 30 sec. The percentage water gained was calculated [27]:

$$\text{Water gained by biochar briquette (\%)} = \frac{w_2 - w_1}{w_1} \times 100 \tag{2}$$

Where: $w_1$ = Initial mass of biochar briquette (g),
$w_2$ = Final mass of biochar briquette (g).

The Water Resistance Capacity (WRC) was determined as [27]:

$$WRC = 100 - [\% \text{ Water gained by briquette}] \qquad (3)$$

## 2.4. Durability of T. cacao pod husk and A. boonei sawdust biochar briquettes

**2.4.1. Fracture toughness.** The ability to resist and withstand an amount of load was assessed for each biochar briquette [28]. Geometric analysis was made on the briquette to determine its width and thickness. In the geometry, a simulated crack was introduced to the briquette by manufacturing a notch with a depth of 0.45 to 0.55 mm. A continuous load was applied to the cracked briquette to determine the critical stress intensity. The load ratio used was R = 0.0258 kN and 15 Hz frequency. The crack opening displacement was recorded using an extensometer mounted at the notch edges created on the briquette. The fracture toughness was determined by this formula [28]:

$$K_I = \frac{4P}{B}\sqrt{\frac{\pi}{W}} \qquad (4)$$

Where: $K_1$ = Fracture toughness or stress intensity (N/cm$^2$).

P = Load applied to the briquette (N).

B = Thickness crack length of the briquette (cm).

W = Width of the briquette (cm).

**2.4.2. Shatter resistance/index.** The resistance to shattering of each biochar briquette was determined using the tumbling test. This involved a cuboid formed by an angle iron frame measuring 16 cm high and 6 cm wide. A sample of each briquette (45 g) inside the cuboid was rotated for about 15 min after which the percentage mass loss of the briquette was calculated [29]:

$$\text{Mass loss (\%)} = \frac{w_1 - w_2}{w_1} \times 100 \qquad (5)$$

Where: $w_1$ = Mass of biochar briquette before shattering (g),

$w_2$ = Mass of biochar briquette after shattering (g).

The Shatter Resistance (%) of each biochar briquette was calculated as [29]:

$$100 - \text{mass loss (\%)} \qquad (6)$$

## 2.5 Proximate properties of T. cacao pod husk and A. boonei sawdust biochar briquettes

**2.5.1 Volatile matter.** Powered samples (2 g with particle size ≤425 μm) from the biochar briquettes from each timber were each placed in a porcelain crucible, oven-dried at 105±2˚C to constant masses, and kept in an electric furnace at 550˚C for 10 min. It was weighed after cooling in a desiccator. The volatile matter for each sample was calculated [30]:

$$\textit{Volatile matter (\%)} = \frac{(B - C)}{B} \times 100 \qquad (7)$$

Where: B = Mass of biochar briquette sample after drying at 105 ± 2˚C (g).

C = Mass of biochar briquette sample after drying at 550˚C (g).

**2.5.2. Ash content.** The ash content of the carbonized wood (2 g) from each biochar briquette was determined using the oven-dried samples (with particle size ≤425 μm) heated in an electric furnace at 600˚C for 4 h. The sample was then cooled in a desiccator, weighed, and the

ash content calculated [30]:

$$Ash\ content\ (\%) = \frac{D}{B} \times 100 \tag{8}$$

where: D = Biochar briquette after drying at 105±2°C (g).

B = Biochar briquette sample after drying at 600°C (g).

**2.5.3. Fixed carbon content.**   The Fixed Carbon Content (FCC) was calculated by subtracting the MC (%), Volatile Matter and ash content (%) of the biochar briquettes from 100 [31].

$$Thus,\ FCC\ (\%) = 100 - (\%Ash + \%Volatile\ matter + \%MC) \tag{9}$$

## 2.6. Combustion properties of T. cacao pod husk and A. boonei sawdust biochar briquettes

**2.6.1. Gas emission analysis.**   Emissions from the combustion of the briquettes were analysed with the use of an Indoor Air Pollution Meter (5000 Series) [32]. Carbon monoxide (CO), one of the most harmful gases to health [32], was analysed. Each briquette (60 g) was arranged on a Biomass Cooking Stove and ignited. The Pollution Meter was hung 1.3 m from the biomass stove and 1.3 m up from the floor in the laboratory to monitor the emissions. It was switched on for a background check for 30 min. before the briquettes were each ignited. After combustion, the Meter was turned off and the time recorded, while a Secure Digital (SD) card in the indoor of the Metre stored the data [32]. The Gas Emission data were compared with the WHO's [33] Indoor Air Quality Standards (i.e., 50 µgm⁻³ for avearage particulate matter and 6 ppm for CO all in 24 h means) to determine their safety to human health and/or the environment.

**2.6.2. Water boiling tests: Burning rate and specific fuel consumption.**   The water boiling test was used to determine the energy transfer capacity of each biochar briquette by assessing their burning rate, Specific Fuel Consumption (SFC), and heat output [34, 35]. A silver beaker of known weight was filled with 100 ml (i.e., 0.1 l) of water, and biochar briquette (100 g) was placed in a biomass cookstove at ambient room temperature and air velocity of 25°C and 0.8 m/s respectively. The temperature of the water in the beaker was measured with a thermocouple and the briquette in the cookstove ignited, while a stopwatch was started. The thermocouple reading was constantly checked at a 5 min.-interval. At 100°C, the beaker was removed from the cookstove and allowed to cool. The remaining amount of water in the beaker was weighed. The briquette fire was quenched and the remaining briquette weighed. The burning rate was estimated by comparing the mass of the briquette burnt (g) with the total time (min.) taken, while the SFC was determined by the mass of the briquette (g) required to boil one liter of water [36].

$$Burning\ rate = \frac{Total\ mass\ of\ burnt\ briquette\ (g)}{Total\ time\ taken\ (min.)} \tag{10}$$

$$Specific\ Fuel\ Consumption = \frac{Mass\ of\ briquette\ consumed\ (g)}{Boiling\ water\ (l)} \tag{11}$$

**2.6.3 Calorific value.**   The Calorific Value (CV) was determined using the Sundy Bomb Calorimeter. Each biochar briquette (1 g) was placed in a crucible. The firing wire was connected to the two electrode rods, which touched the sample in the crucible. Distilled water (10 ml) was poured into the oxygen bomb, the sample was placed into it and the lid of the bomb

tightly closed. The bomb was filled with oxygen at the pressure range of 2.5–4.0 MPa for 10 s after which the pressure valve was released. The bomb was then transferred into the bomb calorimeter. The CV of each biochar briquette was displayed on the computer screen attached to the device [37]

## 2.7. Data analysis

The physical, mechanical and combustion properties were analyzed using Excel 2019. Analysis of Variance (ANOVA) was conducted at 5% probability level. When the differences were significant (p<0.05), Tukey's Multiple Comparison tests were performed (at 5% probability level) using the GraphPad Prism (Version 8) Software.

## 3. Results and discussion

### 3.1. Effect of binder type on the physical properties of T. cacao pod husk and A. boonei sawdust briquettes

**3.1.1 Moisture content.** The MC of the briquettes (Figs 1A–1D and 2A–2D) ranged from 1.7% for the *A. boonei* sawdust/wax biochar briquette to 14.36% for the binderless *T. cacao* pod husk biochar briquette (Fig 3). The low MC for the wax bindered biochar briquettes could be linked to the hydrophilic nature of wax. Generally, it was observed that *T. cacao* pod husk biochar briquettes recorded greater MCs than the *A. boonei* sawdust briquettes irrespective of the binders used. Briquette with great MC is difficult to ignite, and burn with low calorific value, as the water would have to be evaporated out [38]. Thus, binderless *T. cacao* pod husk and *A. boonei* sawdust/starch biochar briquettes with the greatest MCs would be very difficult to ignite. However, the results revealed that only *A. boonei* sawdust/wax, *A. boonei* sawdust/clay, and binderless *A. boonei* sawdust briquettes (with MCs 1.7%, 2.42%, and 4.7% respectively) met the recommended value of <8% by EN 1860 [39] for the European market.

**3.1.2. Basic density.** Basic density gives the integrity of briquette compactness when it falls in water or under the threat of physical and chemical environments [40]. The basic

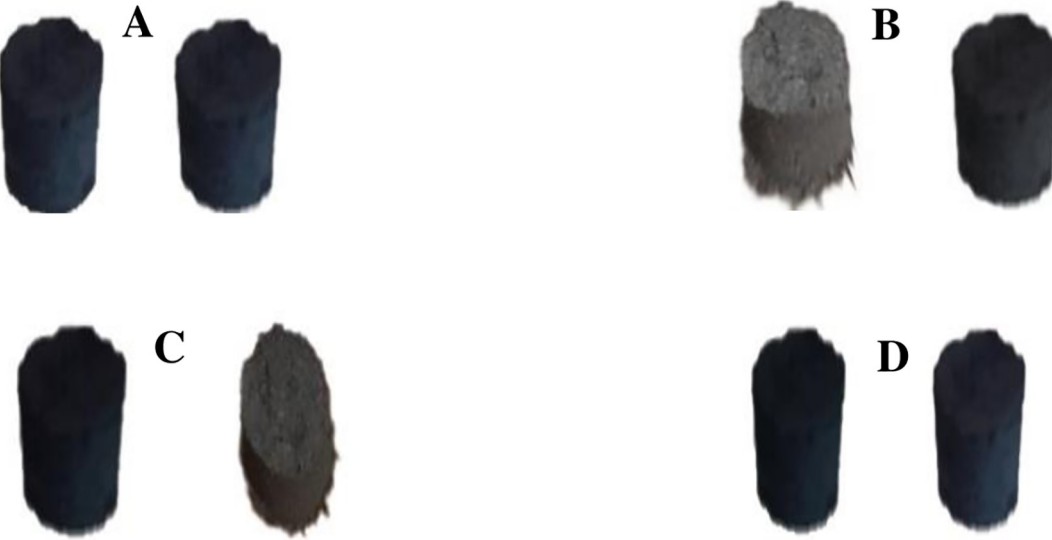

**Fig 1. Charred *T. cacao* pod briquettes.** (A) = wax bindered *T. cacao* pod briquettes, (B) = starch bindered *T. cacao* pod briquettes, (C) = clay bindered *T. cacao* pod briquettes, and (D) = binderless *T. cacao* pod briquettes.

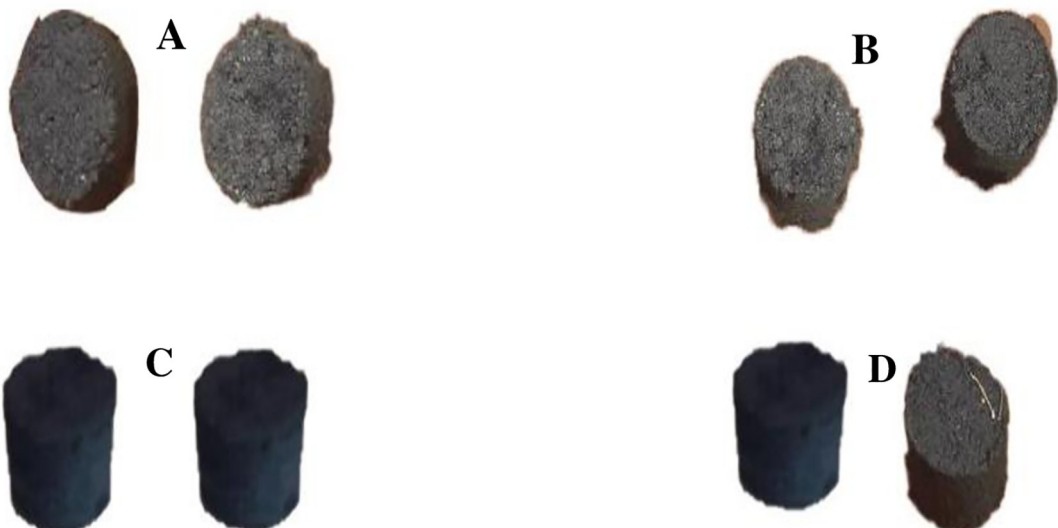

**Fig 2. Charred *A. boonei* sawdust briquettes.** (A) = wax bindered *A. boonei* sawdust briquettes, (B) = starch bindered *A. boonei* sawdust briquettes, (C) = clay bindered *A. boonei* sawdust briquettes, and (D) = binderless *A. boonei* sawdust briquettes.

densities for the briquettes ranged from 279 kg/cm$^3$ (for sawdust/starch) to 433 kg/cm$^3$ (for sawdust/clay) (Fig 4). The differences were significant (p<0.05) (Table 3). The non-bindered *T. cacao* pod briquette was also greater in bulk density (490 kg/cm$^3$) than that of the non-bindered sawdust briquette (328 kg/cm$^3$). The great basic density recorded by the clay briquettes could be attributed to the clay's particle composition. Clay has smaller particle sizes and lower permeability, which made its briquettes bond typically together to create a harder mass than the sawdust's [41]. This sustains the durability, burning rate, and heat capacity of its briquettes. Thus, clay was the most efficient binder in enhancing the basic densities of the briquettes. Binderless or non-bindered *T. cacao* pod briquette also recorded a greater basic density (315 kg/cm$^3$) than its

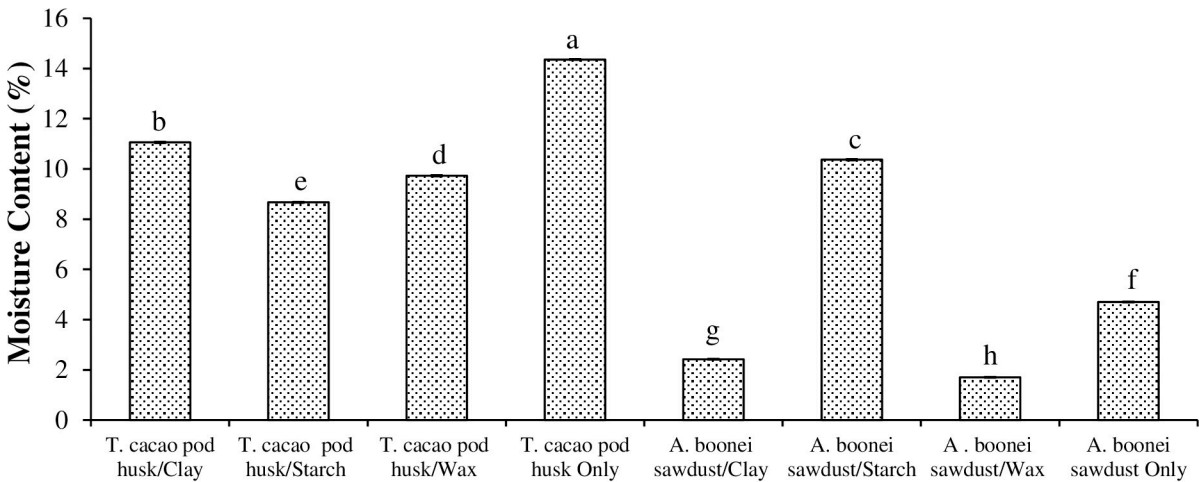

**Bindered and binderless biochar briquettes**

**Fig 3. Moisture content of *T. cacao* pod husk and *A. boonei* sawdust biochar briquettes.** NB: Bars (SE)/Means with the same numerals are not significantly different (p>0.05).

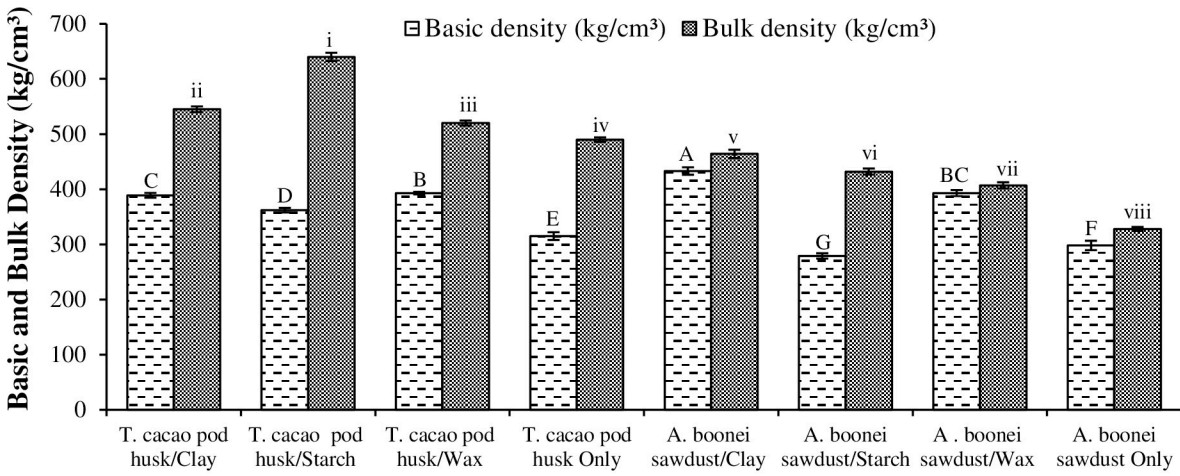

**Fig 4. Basic and bulk densities of *T. cacao* pod husk and *A. boonei* sawdust biochar briquettes.** NB: Bars (SE)/Means with the same letter(s) and numerals for each property are not significantly different (p>0.05).

binderless sawdust counterpart (298 kg/cm³). The *T. cacao* pod residue was generally heavier than the sawdust. This tended to improve the densities of the former's briquettes. Thus, during transport, storage and use, non-bindered *T. cacao* pod briquette would be more stable and energy efficient than the non-bindered *A. boonei* sawdust briquette. Bindered wax briquettes were heavy and equally stable, due to the components of the wax including fatty acid molecules unlike the bindered starch counterparts, which were light and easily dissolved in water during the experimentation process.

**3.1.3. Bulk density.** Bulk density was greatest for the briquettes produced from *T. cacao* pod/starch (640 kg/cm³) and the least for the non-bindered sawdust briquette (328 kg/cm³). Generally, the bulk density for each briquette was greater than its basic density (Fig 4), as the former measured the bulk volume, while the basic density measured the green volume. The differences in the bulk density for the briquettes were also significant (p<0.05) (Table 3). The bulk volume is needed for the energy value of the briquette, while the green volume also helps to identify its density sustainability when exposed to water or moist environment [40]. Briquetting often increases the density of the biomass and improves its handling characteristics. *T. cacao* pod/starch briquette had the greatest bulk density (640 kg/cm³). Such a great bulk density would influence directly the cost of feedstock delivered to a bio-refinery and the storage cost. Onuegbu et al. [36] produced briquettes (with 7 MPa) from elephant grass with bulk density of 319–367 kg/cm³, which is less than those obtained for the two residues (i.e., *T. cacao* pod and *A. boonei* sawdust) under the current investigation. This could be due to the heavier nature of the present feedstocks and the greater compacting pressure (9 MPa) used on them. Starch impacted the greatest bulk density to the *T. cacao* pod briquettes because of its superior adhesion and expansion properties, which tended to hold the cocoa pod particles tightly together. Moreover, it also improved the bulk density of the sawdust briquettes, as its small pore sizes reduced their total pore spaces [42]. Non-bindered *T. cacao* pod briquette had greater bulk density (490 kg/cm³) than the non-bindered sawdust (328 kg/cm³). The finer particles of the cocoa pod bonded better and became more stable than the sawdust [43]. Bulk density is an important indicator in briquetting such that the greater the bulk density, the greater the briquette volume ratio, which

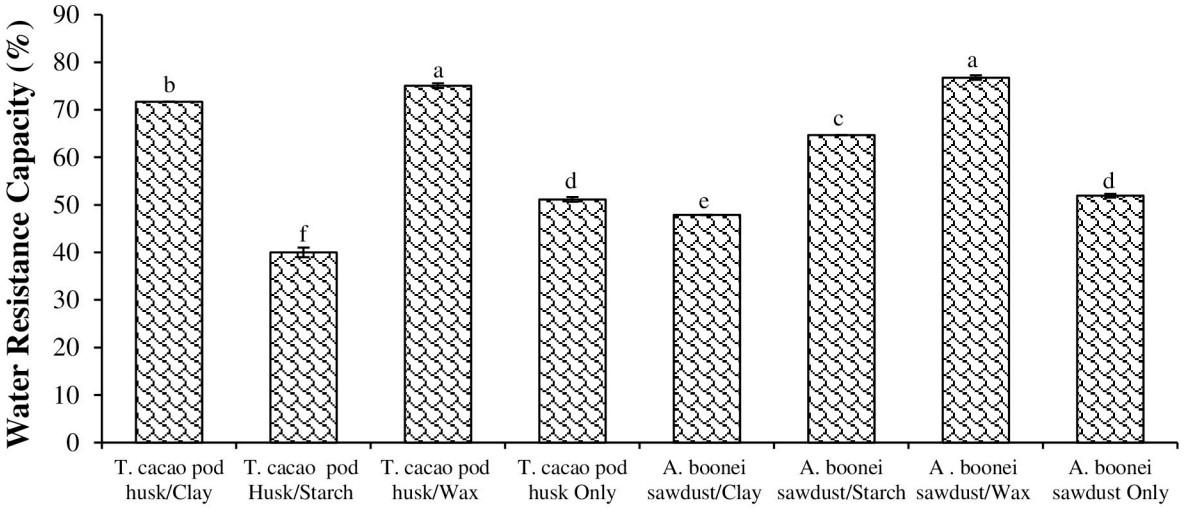

**Fig 5. Water resistance capacity of *T. cacao* pod husk and *A. boonei* sawdust biochar briquettes.** NB: Bars (SE)/Means with the same letter(s) are not significantly different (p>0.05).

is desirable in its transportation, storage, and handling properties. Briquettes with greater bulk densities (e.g., those of *T. cacao* pod/starch) are efficiently suitable for industrial heating.

**3.1.4. Water resistance capacity.** The Water Resistance Capacity (WRC) measured each briquette's ability to withstand prolonged exposure to water [27]. WRC was greatest for *A. boonei* sawdust/wax briquettes (76.76%), as wax (a water-repellent material), which contains long-chain alcohol molecules (about 12–32 carbon atoms) and many fatty acid molecules expanded to seal the pore spaces, which made the briquette withstand water ingress [40]. *T. cacao* pod/starch briquette recorded the least (40.00%) (Fig 5). The differences in WRC between the bindered and the binderless briquettes were significant (p<0.05) (Table 3). Davies et al. [44] observed water resistance values for briquettes pressed under 14 MPa from mango and subabul leaves, and hardwood sawdust to range from 52 to 97.1%, which are much greater than those for the *T. cacao* pod husk and *A. boonei* sawdust briquettes (i.e., 40 to 76.76%) compacted under 9 MPa. Sengar et al. [27] similarly observed that the greater the compaction pressure, the greater the WRC. Wax also enhanced the WRC for its briquettes most due to its insolubility in water. Nonetheless, the non-bindered sawdust briquette also had a greater WRC (51.92%) than its binderless *T. cacao* pod counterpart (51.14%) (Fig 5). This could be due to the fixed geometry and stability of the former after compaction [25, 44], which also influenced well its handling characteristics. This implies that briquetting technologies should aim at achieving great briquette stability for efficiency.

## 3.2. Effect of binder type on the durability of T. cacao pod husk and A. boonei sawdust biochar briquettes

**3.2.1. Fracture toughness.** Fracture toughness describes the ability of a briquette containing a crack to resist breakage [28], and it determines its structural integrity during handling, transportation, and storage when there is breakage. Fig 6 shows that fracture toughness was greatest and least respectively for *T. cacao* pod/clay (i.e., 68.30 N/cm$^2$) and binderless *T. cacao* pod briquettes (i.e., 13.79 N/cm$^2$). Thus, *T. cacao* pod/clay briquette would fracture less during

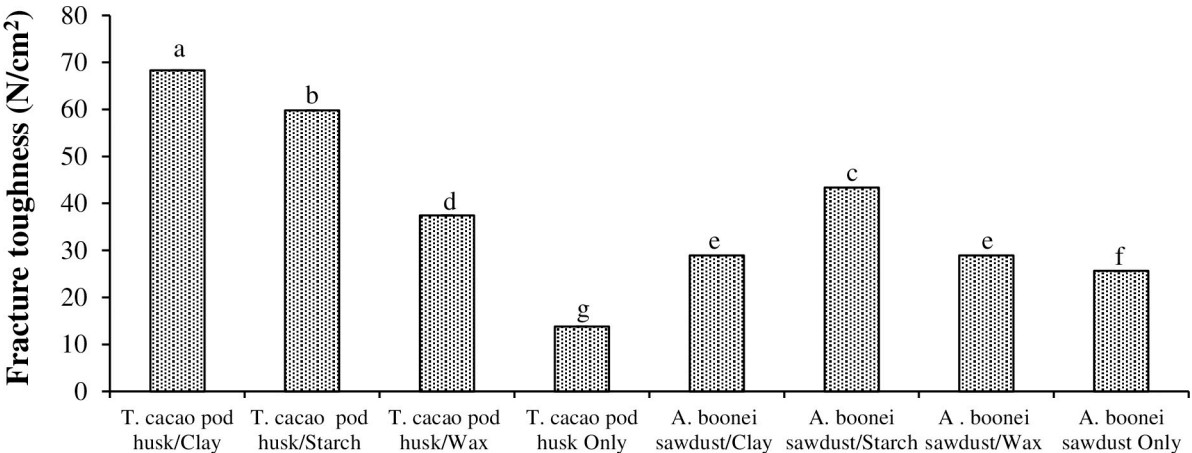

**Fig 6. Fracture toughness of biochar briquettes produced from *T. cacao* pod husk and *A. boonei* sawdust.** NB: Bars (SE)/Means with the same letter(s) are not significantly different (p>0.05).

handling, as the clay dried fastest. As binders, starch followed by clay impacted much toughness efficiently to their briquettes. The small particle size of clay also promoted better bonding and ensured the structural integrity of its briquettes. Non-bindered sawdust briquette also had greater fracture toughness (25.62 N/cm$^2$) than the binderless *T. cacao* pod briquette (13.79 N/cm$^2$). This could be attributed to greater cellulose and lignin in the non-bindered sawdust briquette than the non-bindered *T. cacao* pod briquette, which made the former more compact and stronger. Thus, the non-bindered sawdust briquettes would fracture less under loading.

**3.2.2. Shatter index.** Great Shatter Index (SI) of briquettes is essentially a measure of their resistance to damages from handling, transportation and storage [29]. It contributes in determining their durability. *T. cacao* pod/clay briquette recorded the greatest SI (99.38%) (Fig 7) and resisted shatter most due to its great binding capacity. Starch was most efficient in imparting great shatter resistance to the sawdust briquette because of its bond strength and relative

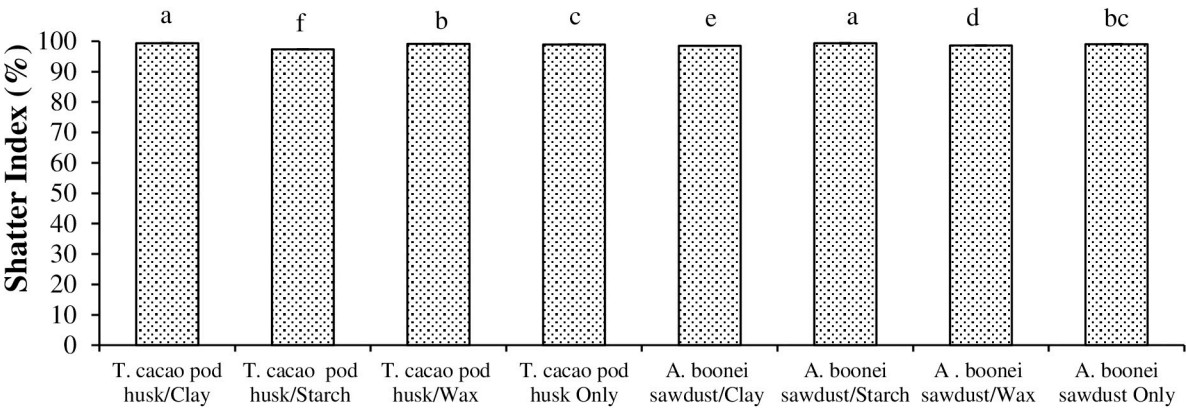

**Fig 7. Shatter Index for biochar briquettes produced from *T. cacao* pod husk and *A. boonei* sawdust.** NB: Bars (SE)/Means with the same letter(s) are not significantly different (p>0.05).

stability. Non-bindered sawdust briquette also resisted shatter (99.04%) more than its binder-less *T. cacao* pod counterpart (98.97%), which was less dense [43]. SI mostly resulted from the uniformity of the biomass/binder mix, density and MC of the briquette [34]. Thus, briquettes from *T. cacao* pod/clay (Shatter Index = 99.38%) and sawdust/starch (Shatter Index = 99.33%) would show great resistance even if handling or storage conditions degenerate. The differences were significant (p<0.05) (Table 3).

### 3.3. Effect of binder type on the proximate properties of T. cacao pod husk and A. boonei sawdust biochar briquettes

**3.3.1 Volatile matter.** The least (21.35%) and greatest (46.41%) volatile matter content were recorded by the *T. cacao* pod husk/clay and *T. cacao* pod husk/Wax biochar briquettes respectively (Fig 8). The result revealed the volatile matter of the biochar briquettes were significantly influenced by the binder type, as there were significant differences (p<0.05) in their volatile matter (Fig 8; Table 3). High volatile matter content is an indication that the briquette would readily ignite with a high proportionate flame during combustion [45]. Briquettes with high volatile matter however burn quite rapidly, and produce a lot of smoke during combustion.

**3.3.2 Ash content.** The non-combustible component obtained from biomass is ash. The ash content for the biochar briquettes investigated varied significantly (p<0.05) (Table 3). Fig 8 shows that *A. boonei* sawdust/clay and binderless *T. cacao* pod husk biochar briquettes recorded the greatest (13.82%) and least (5.19%) for both charred biomasses. The ash content in terms of binder effect however, decreased as clay > starch > wax. The great ash content in the clay bindered biochar briquettes can be attributed to clay's nature as an organic mass with a lot of incombustible minerals. Ash content has a significant effect on heat transfer, oxygen diffusion, to the surface of fuel during combustion and CV [45]. Therefore, the excessive ash content of solid fuel is detrimental to its proper combustion. Hence, the clay bindered *T. cacao* pod husk and *A. boonei* sawdust biochar briquettes would perform poorly based on their high ash contents.

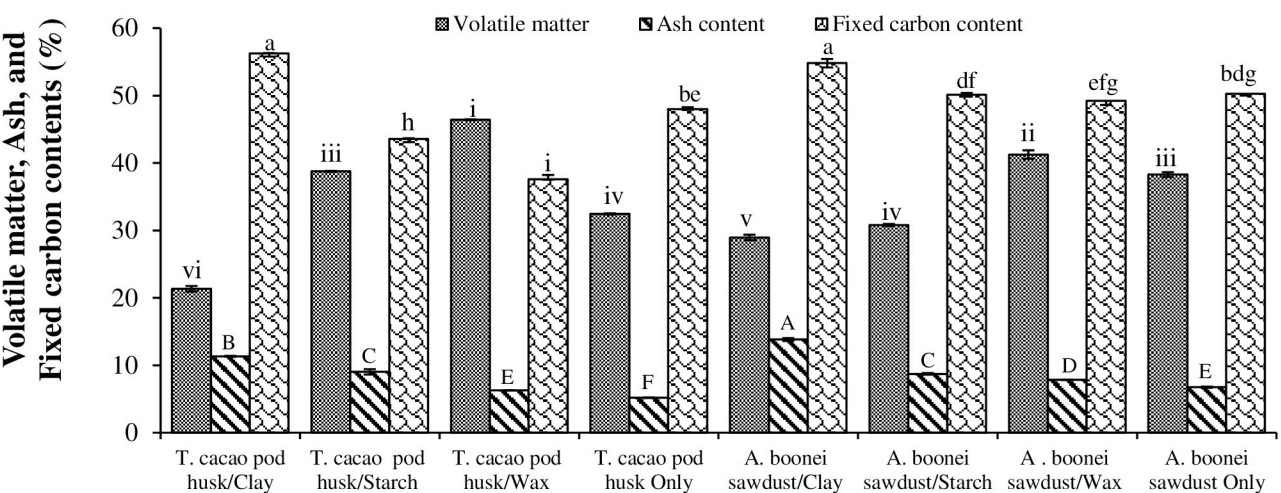

**Fig 8. Proximate properties of biochar briquettes produced from *T. cacao* pod husk and *A. boonei* sawdust.** NB: Bars (SE)/Means with the same letter(s) or numerals for each property are not significantly different (p>0.05).

**3.3.3 fixed carbon content.** Fixed carbon content (FCC) refers to the percentage of carbon available in a biochar for combustion. The FC for the biochar briquettes ranged from 37.59% (for *T. cacao* pod husk/wax) to 56.25% (for *T. cacao* pod husk/clay) (Fig 8) with significant differences ($p < 0.05$) (Table 3). The biochar briquettes being investigated would not be suitable for industrial purposes on the European market, as their FCCs are below the recommended >75% by EN 1860 [39]. The bindered *A. boonei* biochar briquettes recorded higher FCCs than the *T. cacao* counterparts, but for the *T. cacao* pod husk/clay biochar briquette. High percentage of FCC is an indication of a great CV, however other factors like MC, ash, and volatile matter content greatly affect CV.

## 3.4. Effect of binder type on the combustion properties of T. cacao pod husk and A. boonei sawdust biochar briquettes

The Gas Emission Analysis determines the suitability of biochar briquettes to the environment and human health when they are used indoors as fuel [46]. It focused mainly on the average particulate matter ($PM_{2.5}$ in $\mu g/m^3$) and carbon monoxide (CO in ppm) emitted during combustion, which were compared with the WHO [33] Air Quality Guideline Specifications (i.e., $\leq 50\ \mu g m^{-3}$) for particulate matter ($PM_{2.5}$) and $\leq 6$ ppm for CO in 24 h means). Significant differences ($p < 0.05$) in CO and $PM_{2.5}$ emissions for the briquettes during combustion (Table 3) could be due to the physical nature and the chemistry of the feedstocks or raw materials.

**3.4.1. Particulate matter.** $PM_{2.5}$ for wax-bindered *T. cacao* pod biochar briquette (3237.12 $\mu g/m^3$, the greatest), sawdust/wax (124.50 $\mu g/m^3$), *T. cacao* pod clay (123.70 $\mu g/m^3$), sawdust/clay (90.22 $\mu g/m^{-3}$) and non-bindered *T. cacao* pod (80.82 $\mu g/m^3$) (Table 2) were above the WHO [33] Air Quality Standard (i.e., $\leq 50\ \mu g/m^3$). Thus, they would be considered harmful to users. Those from *T. cacao* pod/starch (11.40 $\mu g/m^3$), sawdust/starch (20.95 $\mu g/m^3$) and non-bindered sawdust (47.86 $\mu g/m^3$) would be the least harmful and safe for life, accordingly. Starch-bindered biochar briquettes were safe regarding $PM_{2.5}$ emission, according to the classification, because starch could suppress the emission due to its expanded structure when heated [47]. Wax was the least efficient binder, as it increased the rate of $PM_{2.5}$ emission and released great concentrations in a short time. The great $PM_{2.5}$ of the briquettes could be attributed to the type of the raw material, its physical and chemical compositions [48]. Differences in the non-bindered *T. cacao* pod biochar briquette (80.82 $\mu g/m^3$) and its non-bindered *A. boonei* sawdust type (47.86 $\mu g/m^3$) indicated how the type of the feedstock influenced the $PM_{2.5}$ emission, as the volatile matter in *T. cacao* pod has been estimated by Sayigh [6] to be greater than in sawdust. The more the volatile matter, the greater the harmful $PM_{2.5}$ emission, which adversely affects the human lung and heart [46]. A 10 $\mu g/m^3$ increase in indoor $PM_{2.5}$ increases cardiovascular and respiratory mortalities by 0.63% and 0.75% respectively [46].

**3.4.2. Carbon monoxide.** CO is a harmful greenhouse gas, which influences global warming and climate change [49]. It reacts with hydroxyl ($OH^-$) radicals in the atmosphere to reduce their abundance. Secondary harmful substances or gases are also formed when CO reacts with other forms of matter in the atmosphere [33, 49]. For example, with water, it produces carbonic acid. CO values for biochar briquettes from *T. cacao* pod/starch (17.96 ppm) and sawdust/starch (8.20 ppm) would be considered harmful, while those from non-bindereed *T. cacao* pod (5.42 ppm), *T. cacao* pod/clay (5.33 ppm), *T. cacao* pod/wax (5.16 ppm), non-bindereed *A. boonei* sawdust (4.23 ppm), *A. boonei* sawdust/clay (4.10 ppm) and *A. boonei* sawdust/wax (0.67 ppm) are safe according to WHO [33] Air Quality Standard (i.e., $\leq 6$ ppm). Starch contributed to emitting more CO than the two binders (i.e., wax and clay; Table 2) due to its greatest carbon content. This, coupled with those from the biomass samples, would interact with the atmospheric oxygen to produce CO during burning. Binderless *T. cacao* pod

**Table 2. Burning time and gas emission analysis for *T. cacao* pod husk and *A. boonei* sawdust biochar briquettes.**

| Briquette type | Burning Time (min.) | *Mean particulate matter ($\mu g/m^3$) | •Mean CO (ppm) | Mean Temperature (°C) | Mean Humidity (%) |
|---|---|---|---|---|---|
| Cocoa Pod/Clay | 67 | 123.70[b] | 5.33[bc] | 29.99 | 63.23 |
| Cocoa pod/Starch | 38 | 11.40[e] | 17.90[a] | 33.00 | 53.58 |
| Cocoa pod/wax | 59 | 3237.12[a] | 5.16[bc] | 30.90 | 63.48 |
| Cocoa pod only | 49 | 80.82[bcd] | 5.42[bc] | 32.41 | 59.11 |
| Sawdust/Clay | 51 | 90.22[bc] | 4.10[c] | 32.10 | 60.58 |
| Sawdust/Starch | 54 | 20.95[de] | 8.20[b] | 30.90 | 54.20 |
| Sawdust/Wax | 22 | 124.50[b] | 0.67[d] | 31.19 | 60.11 |
| Sawdust only | 17 | 47.86[cde] | 4.23[c] | 32.00 | 56.72 |

Safe air quality Standard

*Mean particulate matter $\leq$ 50 $\mu g/m^3$

•Mean CO $\leq$ 6 ppm [all in 24 h means] [32].

NB: Means with the same letter(s) as superscripts are not significantly different (p>0.05).

biochar briquette also released more CO than the binderless sawdust sample due to the greater volatile matter in the *T. cacao* pod than in the sawdust [6]. CO readily combines with haemoglobin more than oxygen to form carboxyhemoglobin in humans, which reduces the blood's capacity to transport oxygen [46]. Thus, the time spent during the use of briquettes, which emit much CO during combustion (e.g., that of *T. cacao* pod/starch), especially in closed environments (e.g., kitchens) should be limited. It causes narcosis [46]. Sawdust/wax and sawdust/clay biochar briquettes (0.67 ppm and 4.1 ppm CO respectively) could be used for industrial purposes (e.g., heating) because of their lower emissions and less contribution to global change in climate than the other briquettes under this investigation.

**3.4.3. Water boiling tests: Burning rate and specific fuel consumption.** The burning rate measured the ratio of the mass of the charred briquette sample burnt (g) to the total time (min.) taken [50]. It examined the suitability and the longevity of each briquette under firing [34]. This involves how fast it burns and also releases heat; which are the two combined factors that control the rate of burning [36]. Burning rate for the biochar briquettes ranged from 0.0005 kg/min (for the starch-bindered *A. boonei* sawdust briquette) to 0.0022 kg/min (for sawdust/wax and binderless sawdust briquettes) (Fig 9). Wax also recorded a very quick efficient rate for its briquettes due to its greater melting point, of 49–66°C [51]. Binderless sawdust biochar briquette also had a faster burning rate (0.0022 kg/min) than the binderless *T. cacao* pod biochar briquette (0.0006 kg/min) (Fig 9), as the sawdust is less dense than the *T. cacao* pod [13]. Light biomass has a greater burning rate than heavy ones. Aboagye [48] produced starch-bindered coconut husk briquettes with less burning rate (0.0016–0.0110 kg/min) than those recorded for the briquettes under this study because of the latter's compaction pressure (344.82 kN/m²). The greater the briquette's compaction pressure, the lower its burning rate [52]. As briquettes with slow-burning rates are used in most biomass-fired power plants, the sawdust/clay, sawdust/wax, and non-bindered sawdust biochar briquettes (with 0.0019, 0.0022 kg/min respectively) would be most suitable in power plants.

The Specific Fuel Consumption (SFC) indicated the mass of the briquette (g) required to boil one litre (l) of water [34]. The lower the SFC of the briquette, the more economical it would be. Sawdust/clay biochar briquette recorded the greatest SFC (i.e., 0.2581 kg/l) and sawdust/starch biochar briquette the least (0.0483 kg/l) (Fig 10). Oyelaran [35] recorded the SFC of groundnut shell briquettes with starch (as a binder) to range from 0.067 to 0.267 kg/l. Starch

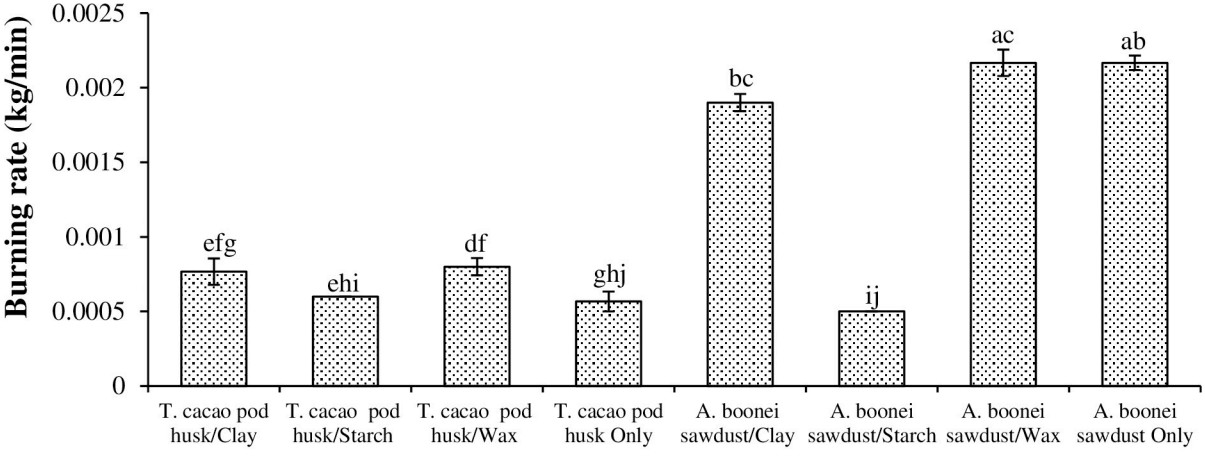

**Fig 9. Burning rate of biochar briquettes produced from *T. cacao* pod husk and *A. boonei* sawdust.** NB: Bars (SE)/Means with the same letter(s) are not significantly different (p>0.05).

was most efficient because it completely and thermally decomposed unlike those bindered with the non-combustible clay. Owing to the light nature of *A. boonei* sawdust, its briquette produced greater SFC (0.2105 kg/l) than that for the non-bindered *T. cacao* pod biochar briquette (0.0934 kg/l), as heavy briquettes possess low SFCs [52]. This implies that the raw cocoa pod could be utilized in the bakery and other related industries where much heat and its retention are a necessary mandate such as in the oven.

**3.4.5. Calorific value.** Fig 11 presents the calorific values (CV) for the biochar briquettes. The clay bindered biochar briquettes binderless *A. boonei* sawdust briquette recorded the greatest (24.733 kJ/kg) CV, while *T. cacao* pod husk/wax biochar briquettes had the lats CV (i.e., 17.150 MJ/kg). Significant differences (p<0.05) (Table 3). The starch bindered biochar briquettes recorded the greatest CVs followed by the clay, and wax bindered briquettes

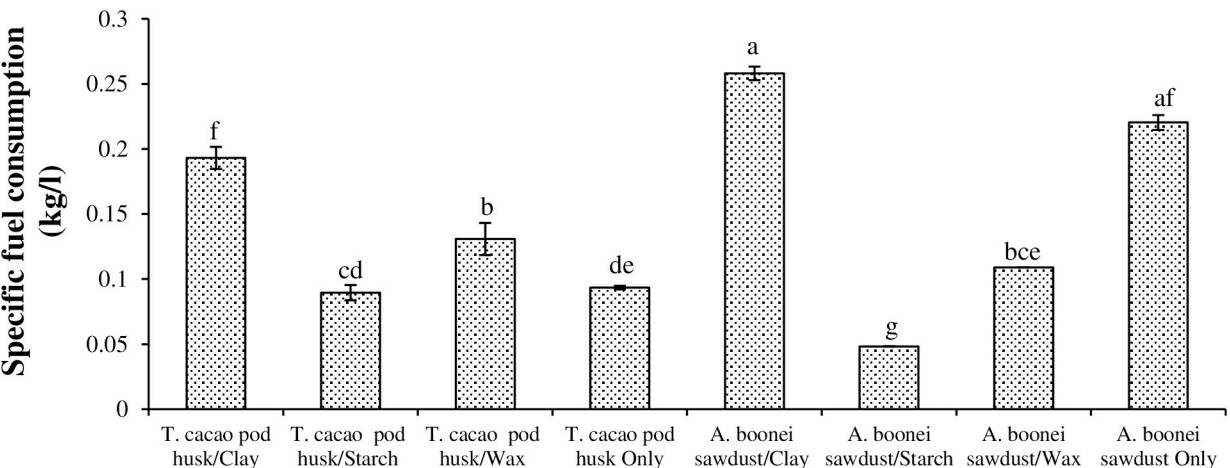

**Fig 10. Specific fuel consumption of biochar briquettes produced from *T. cacao* pod husk and *A. boonei* sawdust.** NB: Bars (SE)/Means with the same letter(s) are not significantly different (p>0.05).

**Table 3. ANOVA for physical, strength and combustion properties of bindered and binderless *T. cacao* pod husk and *A. boonei* sawdust biochar briquettes.**

| Property | Source of variation | df | SS | MS | F | P-value | F critical |
|---|---|---|---|---|---|---|---|
| **Moisture content** | Briquettes | 7 | 421.397 | 60.200 | 4E+05 | 1.86E-40* | **2.6572** |
| | Residual errors | 16 | 0.0026 | 0.0002 | | | |
| | Total | 23 | 421.399 | | | | |
| **Basic density** | Briquettes | 7 | 62224.5 | 8889.2 | 4183.16 | 7.02E-25* | **2.6572** |
| | Residual errors | 16 | 34 | 2.125 | | | |
| | Total | 23 | 62258.5 | | | | |
| **Bulk density** | Briquettes | 7 | 187480.5 | 26782.9 | 7935.68 | 4.19E-27* | **2.6572** |
| | Residual errors | 16 | 54 | 3.375 | | | |
| | Total | 23 | 187534.5 | | | | |
| **Water Resistance Capacity** | Briquettes | 7 | 4070.39 | 581.48 | 1162.79 | 1.94E-20* | **2.6572** |
| | Residual errors | 16 | 8.0012 | 0.500 | | | |
| | Total | 23 | 4078.39 | | | | |
| **Fracture toughness** | Briquettes | 7 | 6978.10 | 996.87 | 6134592 | 3.29E-50* | **2.6572** |
| | Residual errors | 16 | 0.0026 | 0.0002 | | | |
| | Total | 23 | 6978.10 | | | | |
| **Shatter resistance** | Briquettes | 7 | 9.415 | 1.345 | 469.203 | 2.7E-17* | **2.6572** |
| | Residual errors | 16 | 0.046 | 0.0029 | | | |
| | Total | 23 | 9.461 | | | | |
| **Volatile matter content** | Briquettes | 7 | 1321.52 | 188.79 | 550.203 | 7.6E-18* | **2.6572** |
| | Residual errors | 16 | 5.49 | 0.3431 | | | |
| | Total | 23 | 1327.01 | | | | |
| **Ash content** | Briquettes | 7 | 168.15 | 24.021 | 275.51 | 1.8E-15* | **2.6572** |
| | Residual errors | 16 | 1.395 | 0.0872 | | | |
| | Total | 23 | 169.54 | | | | |
| **Fixed Carbon Content** | Briquettes | 7 | 748.13 | 106.9 | 206.03 | 1.8E-14* | **2.6572** |
| | Residual errors | 16 | 8.3 | 0.519 | | | |
| | Total | 23 | 756.4 | | | | |
| **PM$_{2.5}$** | Briquettes | 7 | 2.6E+07 | 4E+06 | 4368.03 | 4.97E-25* | **2.6572** |
| | Residual errors | 16 | 13786.5 | 861.65 | | | |
| | Total | 23 | 2.6E+07 | | | | |
| **CO** | Briquettes | 7 | 545.881 | 77.98 | 45.938 | 2.03E-09 | **2.6572** |
| | Residual errors | 16 | 27.1612 | 1.6976 | | | |
| | Total | 23 | 573.042 | | | | |
| **Burning rate** | Briquettes | 7 | 1.19E-05 | 1.7E-06 | 13.79 | 1.1E-05* | **2.6572** |
| | Residual errors | 16 | 1.97E-06 | 1.2E-07 | | | |
| | Total | 23 | 1.38E-05 | | | | |
| **Specific Fuel Capacity** | Briquettes | 7 | 0.11196 | 0.016 | 132.69 | 5.8E-13* | **2.6572** |
| | Residual errors | 16 | 0.00193 | 0.0001 | | | |
| | Total | 23 | 0.11388 | | | | |
| **Calorific value** | Briquettes | 7 | 129.523 | 18.50 | 26.43 | 1.19E07* | **2.6572** |
| | Residual errors | 16 | 11.2025 | 0.7002 | | | |
| | Total | 23 | 140.726 | | | | |

*Significant differences: $p < 0.05$; F > F critical

(Fig 11). Generally, the greater the CV, the more thermal energy the fuel (i.e., the biochar briquette) would possess. Such biofuel is especially important in high heat demanding applications such as in boilers and steamers. Thus, *T. cacao* pod husk/wax biochar briquettes and *T.*

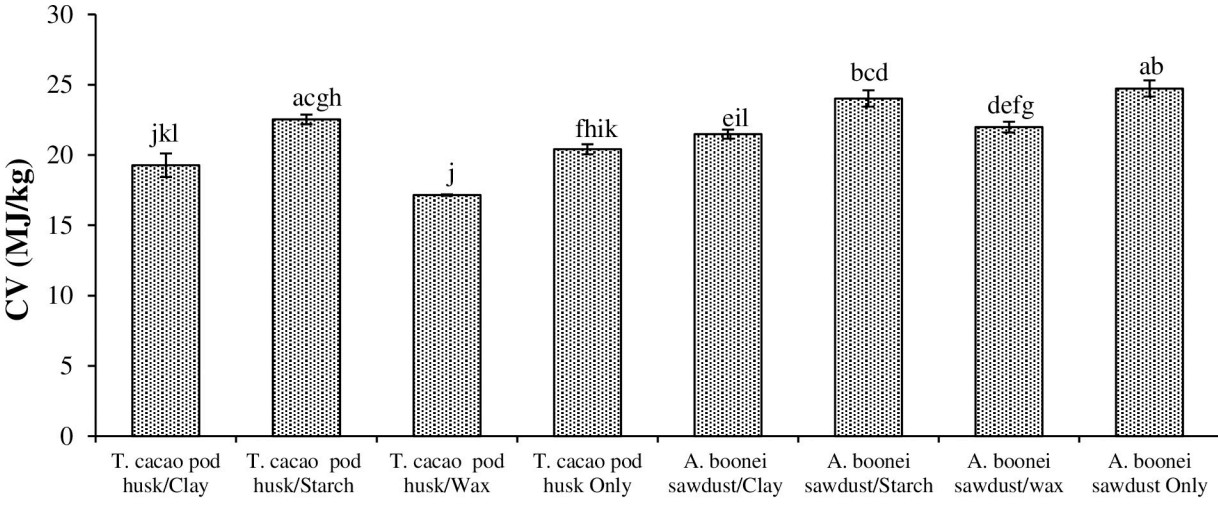

**Fig 11. Calorific value for biochar briquettes produced from *T. cacao* pod husk and *A. boonei* sawdust.** NB: Bars (SE)/Means with the same letter(s) are not significantly different (p>0.05).

*cacao* pod husk/clay briquettes would perform poorly in such applications. CV must however be considered in conjunction with other properties (e.g., MC, density and ash content), which significantly affect briquette quality.

## 4. Conclusion

This study investigated the performance and influence of some binders (i.e., clay, starch, and wax) on the physico-mechanical, proximate, and energy properties of *T. cacao* pod husk and *A. boonei* sawdust biochar briquettes. The findings of the study revealed that, *A. boonei* sawdust/wax briquette recorded the greatest water resistance capacity thus, would resist moisture imbibition most when exposed to the weather. Starch-bindered sawdust briquettes also emitted low particulate matter. Binderless sawdust, sawdust/starch and *T. cacao* pod/starch briquettes recorded 47.86, 20.95 and 11.40 $\mu gm^{-3}$ particulate matter respectively, which are all below the WHO Air Quality Standard and are, thus, safe for domestic uses. *A. boonei* sawdust/starch and *T. cacao* pod/clay briquettes would incur the least handling, transportation and storage fractures due to their great Shatter Indices. As a binder, starch was the most efficient binder. It influenced its briquettes' bulk densities and stabilities. They were the safest regarding particulate matter ($PM_{2.5}$) emission. Clay significantly influenced the basic densities of the briquettes, while wax improved their Water Resistance Capacities the most and produced the least CO. Finally, for binders, which would increase briquette quality, starch is recommended for producing stable biofuels, which are smokeless and environmentally safe. For improved briquette durability, clay proved highly suitable.

## Acknowledgments

We acknowledge the expert advice and technical direction from the German Team (including Messrs. Tene Olivier and Rainer Kuschel, Department of Production and Wood Technology, Technical University of Ostwestfalen-Lippe, Lemgo, Germany), which visited the Department of Wood Science and Technology, KNUST (Kumasi, Ghana) in 2017 on a study tour. We also appreciate the support from the Staff of these KNUST Central Laboratories (i.e., UNDP F and

C), and Mr. Douglas Amoah (Technician, DWST Chemical Laboratory, Faculty of Renewable Natural Resources, KNUST).

## Author Contributions

**Conceptualization:** Charles Antwi-Boasiako, Derrick Adu-Gyamfi.

**Data curation:** Mark Glalah, Derrick Adu-Gyamfi.

**Formal analysis:** Mark Glalah.

**Investigation:** Mark Glalah, Derrick Adu-Gyamfi.

**Methodology:** Mark Glalah, Charles Antwi-Boasiako, Derrick Adu-Gyamfi.

**Project administration:** Charles Antwi-Boasiako.

**Resources:** Charles Antwi-Boasiako, Derrick Adu-Gyamfi.

**Software:** Mark Glalah.

**Supervision:** Charles Antwi-Boasiako.

**Validation:** Charles Antwi-Boasiako.

**Visualization:** Mark Glalah.

**Writing – original draft:** Mark Glalah, Charles Antwi-Boasiako, Derrick Adu-Gyamfi.

**Writing – review & editing:** Mark Glalah, Charles Antwi-Boasiako, Derrick Adu-Gyamfi.

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
