## [Decision Letter · Decision Letter 0]

10 Apr 2023

PONE-D-23-06748Binder-type effect on the physico-mechanical, combustion and emission properties of Alstonia boonei De Wild. sawdust and Theobroma cacao L. pod briquettes as bioenergy productsPLOS ONE

Dear Dr. Antwi-Boasiako (Ph.D),

Thank you for submitting your manuscript to PLOS ONE. After careful consideration, we feel that it has merit but does not fully meet PLOS ONE’s publication criteria as it currently stands. Therefore, we invite you to submit a revised version of the manuscript that addresses the points raised during the review process.

We look forward to receiving your revised manuscript.

Kind regards,

Nakorn Tippayawong, PhD

Academic Editor

PLOS ONE

Journal Requirements:

3. We note that you have referenced (ie. Bewick et al. [5]) which has currently not yet been accepted for publication. Please remove this from your References and amend this to state in the body of your manuscript: (ie “Bewick et al. [Unpublished]”) as detailed online in our guide for authors

Additional Editor Comments (if provided):

1) Please state clearly academic/research contribution of this work to the bioenergy/ agriculture fields, preferably towards the end of the Introduction section

2) All test/ measurement methods used should follow accepted standards. Please refer to the standard methods this work adopted.

Reviewers' comments:

Reviewer's Responses to Questions

**Comments to the Author**

1. Is the manuscript technically sound, and do the data support the conclusions?

Reviewer #1: Partly

Reviewer #2: Yes

2. Has the statistical analysis been performed appropriately and rigorously? 

Reviewer #1: Yes

Reviewer #2: Yes

3. Have the authors made all data underlying the findings in their manuscript fully available?

Reviewer #1: Yes

Reviewer #2: Yes

4. Is the manuscript presented in an intelligible fashion and written in standard English?

Reviewer #1: Yes

Reviewer #2: Yes

5. Review Comments to the Author

Reviewer #1: Comments for the manuscript entitled “Binder-type effect on the physico-mechanical, combustion and emission properties of Alstonia boonei De Wild. sawdust and Theobroma cacao L. pod briquettes as bioenergy products” (MS ID: PONE-D-23-06748)

It is interesting to study the briquetization and characterization of biochar briquette produced from Alstonia boonei De Wild. sawdust and Theobroma cacao L. pod with different binders. However, some aspects are not clear. Thus, the comments for improving this manuscript are listed as below.

Title

- The authors should consider this modified title “Binder-type effect on the physico-mechanical, combustion and emission properties of Alstonia boonei De Wild. and Theobroma cacao L. pod biochar briquettes for energy applications”.

- The title should be clearly indicated as biomass briquettes or biochar briquettes.

Abstract

- Try to avoid wording “Energy production and energy generation”. This is because energy can not be generated following the first law of Thermodynamics. If energy can be produced or generated, it will destroy that law. You may change as “energy application….”.

- Type of combustor and combustion condition strongly affect the emissions both flue gas components and PM concentration, thus this information should be mentioned.

- Key results like energy properties should be reported in this section.

Introduction

- The reason for why biochar briquette is more interesting than biomass briquette should be added in this section.

- The order of reason or research motivation of this section should be revised. The logical order is required of this section.

- The key properties of biochar briquette should be mentioned and which factors affect the biochar briquette properties. Many previous studies have done and reported about the biochar production and characterization.

- The reason (s) for why binder type strongly affects the biochar briquette properties both mechanical and energy aspects, as well as the criteria for selecting the biochar type.

Materials and methods

- The temperature for biochar production needs to be clearly indicated.

- The authors should clearly indicate that this study uses biomass sawdust or biochar sawdust for briquetization.

- The determination of bulk density by water displacement is not suitable for biochar briquette. This is because this material absorbs water during dipping or soaking. Thus, the recorded volume or mass will be error. The non-wetting fluids are more suitable or the way of solid displacement by using very fine grass bead.

- The properties of raw materials should be determined and reported.

- What is the basic density? Normally, the density of materials can be reported as bulk density or true density. It is the density of single briquette or briquettes, this requires the information.

- The authors did not determine the energy properties like gross components (proximate analysis), elemental components (ultimate analysis), heating value and energy density of the produced biochar briquette. These basic properties should be determined and reported.

- The burning test of biochar briquette performed at ambient air or suppling air. The condition should be clearly indicated because it strongly affects the burning rate.

- Do the authors dry the biochar briquette sample after briquetization before determining the properties? The moisture content of dried biochar briquette should be reported.

- For biochar briquetization with binder, the temperature during the process is important for the adhesion of binder and biochar particle. This study did not mention about this issue.

Results and discussion

- The authors need to check the unit of burning rate. The results showed that the value was in the range of 0.5-2.5 x 103 kg/min (Fig. 5), which is so high.

- This study produced solid biofuel but the energy properties of the product are not reported. Thus, the authors should report them.

- The authors reported the specific fuel consumption for boiling water. However, this point may not reply the useful energy of the product. This is because heat loss may be prominent during burning of biochar briquette.

- The authors should give the reasons how binder type and biomass type affect the emission both CO and PM.

- The key results should be clearly indicated and discussed.

- Check the unit of density, for example, 490 kg/cm3 or it is 490 kg/m3?

English

- Carefully check the typing and format of typing. Some sentences have words with capital letter.

Reviewer #2: The manuscript is well-prepared but significant improvements are necessary. Major revision is required. Authors can find the comments as follows:

1) Section 2.1: Carbonization temperature and heating rate should be reported.

2) Citation is very excessive. In general, original research article should have not more than 50 references. Please recheck and remove some references that are not related to the text. For example, citation [43] has nothing to do with the basic density definition or method. Only reference [42] is enough. Also, reference [40] may not be necessary, instead the detail method of quenching and weighing can be described.

3) It is better (if possible) to have real evidence by performing some analysis work instead of just citing the reference. For example, the structure or constituent of each binder should be analyzed so their characteristics that affect the property of the briquette can be investigated. For example, Page 11 Line 259-260, fatty acid molecule in wax gave the resulted briquette heavy and stable unlike the one made with starch. Also, authors discuss about pore size of the material. It is better to perform BET surface analysis of the samples instead of just citing the reference (Page 12, Line 281-282).

4) Figure 3 is strange. density values should not be presented with the water resistance capacity. They have different unit and meaning. Also, their values are significantly different; the scale should be adjusted appropriately. Figure 4 and 5 has the same issue.

5) Fracture toughness results are very interesting and unique. However, the discussion on the effect of pore size of clay is based on citation (Page 15, Line 336). It is better to have real pore size analysis and discuss with your own results. Also, Line 335 read "the lower the MC of the briquette, the greater is its fracture toughness [27]", is that your own results or it is only from the citation? Please clarify. If MC is the parameter during briquetting process, it should be mentioned in the methodology.

6) What is a,b,c,d,e superscript in Table 2.

7) Conclusion should be rewritten. Some opening sentences are needed before bullet conclusions.

6. PLOS authors have the option to publish the peer review history of their article (what does this mean?). If published, this will include your full peer review and any attached files.

Reviewer #1: No

Reviewer #2: No

---

## [Author Response · Author response to Decision Letter 0]

2 Nov 2023

Reviewer #1:

Comments for the manuscript entitled “Binder-type effect on the physico-mechanical, combustion and emission properties of Alstonia boonei De Wild. sawdust and Theobroma cacao L. pod briquettes as bioenergy products” (MS ID: PONE-D-23-06748)

It is interesting to study the briquetization and characterization of biochar briquette produced from Alstonia boonei De Wild. sawdust and Theobroma cacao L. pod with different binders. However, some aspects are not clear. Thus, the comments for improving this manuscript are listed as below.

Title

- The authors should consider this modified title “Binder-type effect on the physico-mechanical, combustion and emission properties of Alstonia boonei De Wild. and Theobroma cacao L. pod biochar briquettes for energy applications”.

Authors’ response: This advice is taken and has been effected in the manuscript.

- The title should be clearly indicated as biomass briquettes or biochar briquettes.

Authors’ response: This advice is taken and the title has been re-written to capture ‘biochar briquettes’.

Abstract

- Try to avoid wording “Energy production and energy generation”. This is because energy can not be generated following the first law of Thermodynamics. If energy can be produced or generated, it will destroy that law. You may change as “energy application….”.

Author's response: This advice is taken and duly corrected.

- Type of combustor and combustion condition strongly affect the emissions both flue gas components and PM concentration, thus this information should be mentioned.

Author's response: These have been duly added in the abstract now. PAGE 2, LINE 38 & 39.

- Key results like energy properties should be reported in this section.

Author's response: Such have already been captured and updated. PAGE 2, LINE 40 - 49

Introduction

- The reason for why biochar briquette is more interesting than biomass briquette should be added in this section.

Author's response: These have been duly given now. PAGE 3, LINES 69 – 744; PAGE 4, LINE, 89; PAGE 5, LINE, 115 – 117.

- The order of reason or research motivation of this section should be revised. The logical order is required of this section.

Author's response: These have been considered and duly effected.

- The key properties of biochar briquette should be mentioned and which factors affect the biochar briquette properties. Many previous studies have done and reported about the biochar production and characterization.

Author's response: Such had been given briefly, and a little more added in this review. LINE PAGE 5, 119 - 125

- The reason (s) for why binder type strongly affects the biochar briquette properties both mechanical and energy aspects, as well as the criteria for selecting the biochar type.

Author's response: This has been clearly added now –LINE 121 – 123. The criteria for selecting the biochar type can be found from Line 84 – 100. They were selected based on the large volumes of the waste being produced. 

Materials and methods

- The temperature for biochar production needs to be clearly indicated.

Author's response: This has been clearly added now – LINE 153-154.

- The authors should clearly indicate that this study uses biomass sawdust or biochar sawdust for briquetization.

Author's response: T. cacao pod husk and A. boonei sawdust were carbonized to produce biochar briquettes and this has been clearly stated at all points now.

- The determination of bulk density by water displacement is not suitable for biochar briquette. This is because this material absorbs water during dipping or soaking. Thus, the recorded volume or mass will be error. The non-wetting fluids are more suitable or the way of solid displacement by using very fine grass bead.

Author's response: This was considered during determination and rewritten to capture necessary steps taken to prevent moisture uptake. The biochar briquettes were tightly wrapped in light cling film (Everpack®) (negligible weight) to prevent water imbibition.

- The properties of raw materials should be determined and reported.

Author's response: The study sought to investigate the binder effect on the biochar properties, and not the biomass raw material briquette. With this in mind, the properties of the non-bindered samples (Controls) were determined and data reported.

- What is the basic density? Normally, the density of materials can be reported as bulk density or true density. It is the density of single briquette or briquettes, this requires the information.

Author's response: Basic density is different from bulk density, and averages of all data for the various properties are what is reported. Raw data from replications were used for data analysis.

- The authors did not determine the energy properties like gross components (proximate analysis), elemental components (ultimate analysis), heating value and energy density of the produced biochar briquette. These basic properties should be determined and reported.

Author's response: The authors had already determined energy properties like Specific fuel consumption, and burning rate. We have however now determined the Proximate properties and calorific values of the biochar briquettes and provided reports of them in the review.

- The burning test of biochar briquette performed at ambient air or suppling air. The condition should be clearly indicated because it strongly affects the burning rate.

Author's response: Ambient room temperature and air velocity of 25 ℃ and 0.8 m/s respectively – Line 284 & 285

- Do the authors dry the biochar briquette sample after briquetization before determining the properties? The moisture content of dried biochar briquette should be reported.

Author's response: The moisture content of the biochar briquettes has been reported.

- For biochar briquetization with binder, the temperature during the process is important for the adhesion of binder and biochar particle. This study did not mention about this issue.

Author's response: The clay and starch binders used do not need high or low temperatures to cure, both required just ambient temperature. Only wax paste was prepared by boiling and the temperature of 100℃ has now been indicated - LINE 165 - 167.

Results and discussion

- The authors need to check the unit of burning rate. The results showed that the value was in the range of 0.5-2.5 x 103 kg/min (Fig. 5), which is so high.

Author's response: The values have been checked and scale corrected.

- This study produced solid biofuel but the energy properties of the product are not reported. Thus, the authors should report them.

Author's response: Energy properties; calorific value, proximate properties, burning rate, and specific fuel consumption are reported.

- The authors reported the specific fuel consumption for boiling water. However, this point may not reply the useful energy of the product. This is because heat loss may be prominent during burning of biochar briquette.

Author's response: Their calorific value was reported and burning rate have been reported. The specific fuel consumption for boiling reported is very important not only for water heating but also in applications like cooking, barbecuing, and all such energy applications. The authors employed scientifically published and standard methods in all property determinations.

- The authors should give the reasons how binder type and biomass type affect the emission both CO and PM.

Author's response: This has been done.

- The key results should be clearly indicated and discussed.

Author's response: This has been done.

- Check the unit of density, for example, 490 kg/cm3 or it is 490 kg/m3?

Author's response: They are both S.I units and correct. The authors decided to go with the former.

English

- Carefully check the typing and format of typing. Some sentences have words with capital letter.

Author's response: All such have been done and corrected where necessary.

Reviewer #2:

The manuscript is well-prepared but significant improvements are necessary. Major revision is required. Authors can find the comments as follows:

1) Section 2.1: Carbonization temperature and heating rate should be reported.

Author's response: The carbonization temperature and heating rate (i.e., (at 410 ± 5°C, with a heating rate of 4°C/min from the ambient temperature of 25°C) has been reported now.

2) Citation is very excessive. In general, original research article should have not more than 50 references. Please recheck and remove some references that are not related to the text. For example, citation [43] has nothing to do with the basic density definition or method. Only reference [42] is enough. Also, reference [40] may not be necessary, instead the detail method of quenching and weighing can be described.

Author's response: This has done and the references reduced to 49.

3) It is better (if possible) to have real evidence by performing some analysis work instead of just citing the reference. For example, the structure or constituent of each binder should be analyzed so their characteristics that affect the property of the briquette can be investigated. For example, Page 11 Line 259-260, fatty acid molecule in wax gave the resulted briquette heavy and stable unlike the one made with starch. Also, authors discuss about pore size of the material. It is better to perform BET surface analysis of the samples instead of just citing the reference (Page 12, Line 281-282).

Author's response: The authors are of the view that evidential studies as suggested is apt, however, the references cited from scientific papers are also very much apt. 

4) Figure 3 is strange. density values should not be presented with the water resistance capacity. They have different unit and meaning. Also, their values are significantly different; the scale should be adjusted appropriately. Figure 4 and 5 has the same issue.

Author's response: These figures have all now been represented separately.

5) Fracture toughness results are very interesting and unique. However, the discussion on the effect of pore size of clay is based on citation (Page 15, Line 336). It is better to have real pore size analysis and discuss with your own results. Also, Line 335 read "the lower the MC of the briquette, the greater is its fracture toughness [27]", is that your own results or it is only from the citation? Please clarify. If MC is the parameter during briquetting process, it should be mentioned in the methodology.

Author's response: The authors are of the view that evidential studies as suggested is apt, however, the references cited from scientific papers are also very much apt, thus adopted. 

For the second part, it is from the citation.

For the third part, the MC of the biochar briquettes were determined and now reported accordingly.

6) What is a,b,c,d,e superscript in Table 2.

Author's response: They represent differences or not between the various treatments. “Bars/Means with the same letter(s) and numerals for each property are not significantly different (p>0.05).”

7) The conclusion should be rewritten. Some opening sentences are needed before bullet conclusions.

Author's response: This has been done.

---

## [Decision Letter · Decision Letter 1]

17 Jan 2024

PONE-D-23-06748R1Binder-type effect on the physico-mechanical, combustion and emission properties of Alstonia boonei De Wild. sawdust and Theobroma cacao L. pod biochar briquettes for energy applicationsPLOS ONE

Dear Dr. Antwi-Boasiako (Ph.D),

Thank you for submitting your manuscript to PLOS ONE. After careful consideration, we feel that it has merit but does not fully meet PLOS ONE’s publication criteria as it currently stands. Therefore, we invite you to submit a revised version of the manuscript that addresses the points raised during the review process.

We look forward to receiving your revised manuscript.

Kind regards,

Nakorn Tippayawong, PhD

Academic Editor

PLOS ONE

Journal Requirements:

Reviewers' comments:

Reviewer's Responses to Questions

**Comments to the Author**

1. If the authors have adequately addressed your comments raised in a previous round of review and you feel that this manuscript is now acceptable for publication, you may indicate that here to bypass the “Comments to the Author” section, enter your conflict of interest statement in the “Confidential to Editor” section, and submit your "Accept" recommendation.

Reviewer #1: All comments have been addressed

Reviewer #2: All comments have been addressed

2. Is the manuscript technically sound, and do the data support the conclusions?

Reviewer #1: Partly

Reviewer #2: Yes

3. Has the statistical analysis been performed appropriately and rigorously? 

Reviewer #1: Yes

Reviewer #2: Yes

4. Have the authors made all data underlying the findings in their manuscript fully available?

Reviewer #1: Yes

Reviewer #2: Yes

5. Is the manuscript presented in an intelligible fashion and written in standard English?

Reviewer #1: Yes

Reviewer #2: Yes

6. Review Comments to the Author

Reviewer #1: Comments for the revised manuscript “Binder-type effect on the physico-mechanical, combustion and emission properties of Alstonia boonei De Wild. sawdust and Theobroma cacao L. pod briquettes as bioenergy products” (MS. no: PONE-D-23-06748R1)

- The authors must check the unit of bulk density in the abstract and other sections. How it is possible the bulk density was 640 kg/cm3 or 433 kg/cm3? If you mean 640 g/cm3 or 433 g/cm3, I agree with you for this. It is ok for SI unit, but the unit is not correct?

- The authors should provide the space between number and its unit. Check throughout the manuscript.

- I still do not agree with the basic density, if you mean bulk density, you should use this word. And, you have to indicate that it is bulk density of single briquette, not briquettes to avoid the confusion of the reader.

- Page 10 and line 224: the author needs to check the mass of sample of briquette for boiling test. You used only 0.087 kg for boiling 2 kg of sample (water+beaker), how is it possible?

- Page 10, line 220: the author needs to check this meaning “energy efficiency transfer”, it is suitable for your test?

- For biochar briquetization with binder, the authors did not measure the temperature of cylindrical steel mould (near to inside wall). How the authors know that the binder (starch) is gelatinization for binding the biochar particles?

Reviewer #2: Authors responded and revised manuscript according to the suggestion. The manuscript is ready for publication in the journal.

7. PLOS authors have the option to publish the peer review history of their article (what does this mean?). If published, this will include your full peer review and any attached files.

Reviewer #1: No

Reviewer #2: No

---

## [Author Response · Author response to Decision Letter 1]

26 Jan 2024

Reviewer #1: 

Comments for the revised manuscript “Binder-type effect on the physico-mechanical, combustion and emission properties of Alstonia boonei De Wild. sawdust and Theobroma cacao L. pod briquettes as bioenergy products” (MS. no: PONE-D-23-06748R1)

- The authors must check the unit of bulk density in the abstract and other sections. How it is possible the bulk density was 640 kg/cm3 or 433 kg/cm3? If you mean 640 g/cm3 or 433 g/cm3, I agree with you for this. It is ok for SI unit, but the unit is not correct?

RESPONSE: The authors maintain the values reported are correct. Several research (e.g., [1] and [2] below) have reported similar trends for briquettes.

[Bulk density is the mass per unit volume of a bulk of a material, including the air voids between the particles, and inside the particles. Like the mass of a cubic of crushed chalk. Usually obtained by weighing a container of known volume full of the material after a set vibration and compaction regime, While

Basic density on the other hand measures mass per unit true volume. this volume does not include the pore spaces in between the granules.]

[1] Mibulo, T., Nsubuga, D., Kabenge, I., & Wydra, K. D. (2023). Characterization of briquettes developed from banana peels, pineapple peels and water hyacinth. Energy, Sustainability and Society, 13(1), 36. https://doi.org/10.1186/s13705-023-00414-3

[2] Himbane, P. B., Grand Ndiaye, L., Napoli, A., & Kobor, D. (2018). Physicochemical and mechanical properties of biomass coal briquettes produced by artisanal method. African Journal of Environmental Science and Technology, 12(12), 480-486. DOI: 10.5897/AJEST2018.2568

- The authors should provide the space between number and its unit. Check throughout the manuscript.

RESPONSE: This has been done now in the current submission

- I still do not agree with the basic density, if you mean bulk density, you should use this word. And, you have to indicate that it is bulk density of single briquette, not briquettes to avoid the confusion of the reader.

RESPONSE: Bulk density as used here is not to imply the collective density of that of all the biochar briquettes. Basic and bulk density are different properties assessed.

[Bulk density is the mass per unit volume of a bulk of a material, including the air voids between the particles, and inside the particles. Like the mass of a cubic of crushed rock. Usually obtained by weighing a container of known volume full of the material after a set compaction regime, While

Basic density on the other hand measures mass per unit true volume of the grains. this volume does not include the pore spaces in between the granules.]

- Page 10 and line 224: the author needs to check the mass of sample of briquette for boiling test. You used only 0.087 kg for boiling 2 kg of sample (water+beaker), how is it possible?

RESPONSE: The authors have referred and accordingly corrected the report. Biochar briquette (100 g) and water of 100 ml = 100 cm3 = 0.1 l were used. (PG 10, Line 253 – 255).

- Page 10, line 220: the author needs to check this meaning “energy efficiency transfer”, it is suitable for your test?

RESPONSE: The sentence has been updated in this current submission to convey a clearer understanding. 

- For biochar briquetization with binder, the authors did not measure the temperature of cylindrical steel mould (near to inside wall). How the authors know that the binder (starch) is gelatinization for binding the biochar particles?

RESPONSE: Authors noted that, for briquetization, the commercial starch was prepared by boiling at 100 ℃ making it gelatinous and/or gluey. PG 6, Line 142 -143.

Reviewer #2: 

Authors responded and revised manuscript according to the suggestion. The manuscript is ready for publication in the journal.

---

## [Decision Letter · Decision Letter 2]

31 Jan 2024

PONE-D-23-06748R2Binder-type effect on the physico-mechanical, combustion and emission properties of Alstonia boonei De Wild. sawdust and Theobroma cacao L. pod biochar briquettes for energy applicationsPLOS ONE

Dear Dr. Glalah,

Thank you for submitting your manuscript to PLOS ONE. After careful consideration, we feel that it has merit but does not fully meet PLOS ONE’s publication criteria as it currently stands. Therefore, we invite you to submit a revised version of the manuscript that addresses the points raised during the review process.

We look forward to receiving your revised manuscript.

Kind regards,

Nakorn Tippayawong, PhD

Academic Editor

PLOS ONE

Journal Requirements:

Reviewers' comments:

Reviewer's Responses to Questions

**Comments to the Author**

1. If the authors have adequately addressed your comments raised in a previous round of review and you feel that this manuscript is now acceptable for publication, you may indicate that here to bypass the “Comments to the Author” section, enter your conflict of interest statement in the “Confidential to Editor” section, and submit your "Accept" recommendation.

Reviewer #1: All comments have been addressed

2. Is the manuscript technically sound, and do the data support the conclusions?

Reviewer #1: Yes

3. Has the statistical analysis been performed appropriately and rigorously? 

Reviewer #1: (No Response)

4. Have the authors made all data underlying the findings in their manuscript fully available?

Reviewer #1: Yes

5. Is the manuscript presented in an intelligible fashion and written in standard English?

Reviewer #1: Yes

6. Review Comments to the Author

Reviewer #1: The unit of bulk density is not correct. It must be kg/m3 or g/cm3. The authors must improve this issue throughout the manuscript.

7. PLOS authors have the option to publish the peer review history of their article (what does this mean?). If published, this will include your full peer review and any attached files.

Reviewer #1: No

---

## [Author Response · Author response to Decision Letter 2]

26 Mar 2024

Reviewer #1: The unit of bulk density is not correct. It must be kg/m3 or g/cm3. The authors must improve this issue throughout the manuscript.

Author’s Response: The unit for density has been corrected to kg/m3 throughout the entire manuscript

---

## [Decision Letter · Decision Letter 3]

10 May 2024

PONE-D-23-06748R3Binder-type effect on the physico-mechanical, combustion and emission properties of Alstonia boonei De Wild. sawdust and Theobroma cacao L. pod biochar briquettes for energy applicationsPLOS ONE

Dear Dr. Glalah,

Thank you for submitting your manuscript to PLOS ONE. After careful consideration, we feel that it has merit but does not fully meet PLOS ONE’s publication criteria as it currently stands. Therefore, we invite you to submit a revised version of the manuscript that addresses the points raised during the review process.

Reviewer #3: This manuscript discussed char briquette made of two kinds of raw materials with different binder namely clay, starch, and wax. Some questions should be raised since the experiment on these products are less clear, for instances:

1. Why authors choose char-briquettes instead of briquette? This information should be included in introduction section.

2. Why authors use wax as the binder? It is not safe if the char briquettes are used as barbeques fuel!

3. It is better to include the chemical content of the raw materials (pod of cacao and sawdust of Alstonia wood) as well as after they become char.

4. The composition of mixture pod and sawdust was missing in the method.

5. The binder should be stated as weight/weight (w/w) or weight/volume (w/v).

6. The authors emphasized on binder but the fundamental characteristics of the binders (origin, chemical content, MSDS/ material safety data sheet, etc) were missing in the text.

7. Authors mentioned 10 replications for each type of the briquette. However, when I checked into the excel (supplementary materials), there were only three replications that authors did for testing.

8. Further, for statistical analysis, I could not find the complete analysis. They should be attached in the excel supplementary materials

9. Bio-char is related also as the absorbent instead of only for fuel. Why authors did not test for this purpose?

10. Please make the conclusion concise! Not point by point!

Reviewer #4: The document evaluated is very interesting and is also the result of rigorous and well-cared for research. I would only recommend that you consider your projection of the results of this work in other parts of the world that have these natural resources.

We look forward to receiving your revised manuscript.

Kind regards,

Noé Aguilar-Rivera

Academic Editor

PLOS ONE

---

## [Author Response · Author response to Decision Letter 3]

7 Jun 2024

Reviewer #3: 

This manuscript discussed char briquette made of two kinds of raw materials with different binder namely clay, starch, and wax. Some questions should be raised since the experiment on these products are less clear, for instances:

1. Why authors choose char-briquettes instead of briquette? This information should be included in introduction section.

AUTHOR’S RESPONSE: This has now been included in the Introduction Section; LINE 119-124

2. Why authors use wax as the binder? It is not safe if the char briquettes are used as barbeques fuel!

AUTHOR’S RESPONSE: Wax was analyzed/investigated as a binder because of its ability to bind, NOT necessarily to promote its use as a briquette binder. 

3. It is better to include the chemical content of the raw materials (pod of cacao and sawdust of Alstonia wood) as well as after they become char.

AUTHOR’S RESPONSE: The authors did not include this because, after charring, the primary content of charred biomasses is carbon regardless of the composition of the biomaterial. Other trace elements might be present, but the composition of such does not constitute the objective of this study. 

4. The composition of mixture pod and sawdust was missing in the method.

AUTHOR’S RESPONSE: Table 1 has been updated to carry this information as well.

5. The binder should be stated as weight/weight (w/w) or weight/volume (w/v).

AUTHOR’S RESPONSE: Table 1and the note under has been updated to carry this information.

6. The authors emphasized on binder but the fundamental characteristics of the binders (origin, chemical content, MSDS/ material safety data sheet, etc) were missing in the text.

AUTHOR’S RESPONSE: This has been added to SECTION 2.2

7. Authors mentioned 10 replications for each type of the briquette. However, when I checked into the excel (supplementary materials), there were only three replications that authors did for testing.

AUTHOR’S RESPONSE: Typographical error in reporting, this has been duly rectified.

8. Further, for statistical analysis, I could not find the complete analysis. They should be attached in the excel supplementary materials

AUTHOR’S RESPONSE: Such have been attached.

9. Bio-char is related also as the absorbent instead of only for fuel. Why authors did not test for this purpose?

AUTHOR’S RESPONSE: The authors sought to test their performance as a biomass fuel and not as absorbent materials.

10. Please make the conclusion concise! Not point by point!

AUTHOR’S RESPONSE: This has been addressed.

Reviewer #4: 

The document evaluated is very interesting and is also the result of rigorous and well-cared for research. I would only recommend that you consider your projection of the results of this work in other parts of the world that have these natural resources.

AUTHOR’S RESPONSE: This has been addressed in this conclusion.

---

## [Editor Report · Decision Letter 4]

24 Jun 2024

Binder-type effect on the physico-mechanical, combustion and emission properties of Alstonia boonei De Wild. sawdust and Theobroma cacao L. pod biochar briquettes for energy applications

PONE-D-23-06748R4

Dear Dr. Mark Glalah

We’re pleased to inform you that your manuscript has been judged scientifically suitable for publication and will be formally accepted for publication once it meets all outstanding technical requirements.

Kind regards,

Noé Aguilar-Rivera

Academic Editor

PLOS ONE
---

## [Editor Report · Acceptance letter]

18 Jul 2024

PONE-D-23-06748R4 

PLOS ONE

Dear Dr. Glalah, 

I'm pleased to inform you that your manuscript has been deemed suitable for publication in PLOS ONE. Congratulations! Your manuscript is now being handed over to our production team.

Kind regards, 

on behalf of

Dr. Noé Aguilar-Rivera 

Academic Editor

PLOS ONE